# EVA3D: COMPOSITIONAL 3D HUMAN GENERATION FROM 2D IMAGE COLLECTIONS

**Fangzhou Hong,   Zhaoxi Chen,   Yushi Lan,   Liang Pan,   Ziwei Liu** ✉

S-Lab, Nanyang Technological University

{fangzhou001, zhaoxi001, yushi001, liang.pan, ziwei.liu}@ntu.edu.sg

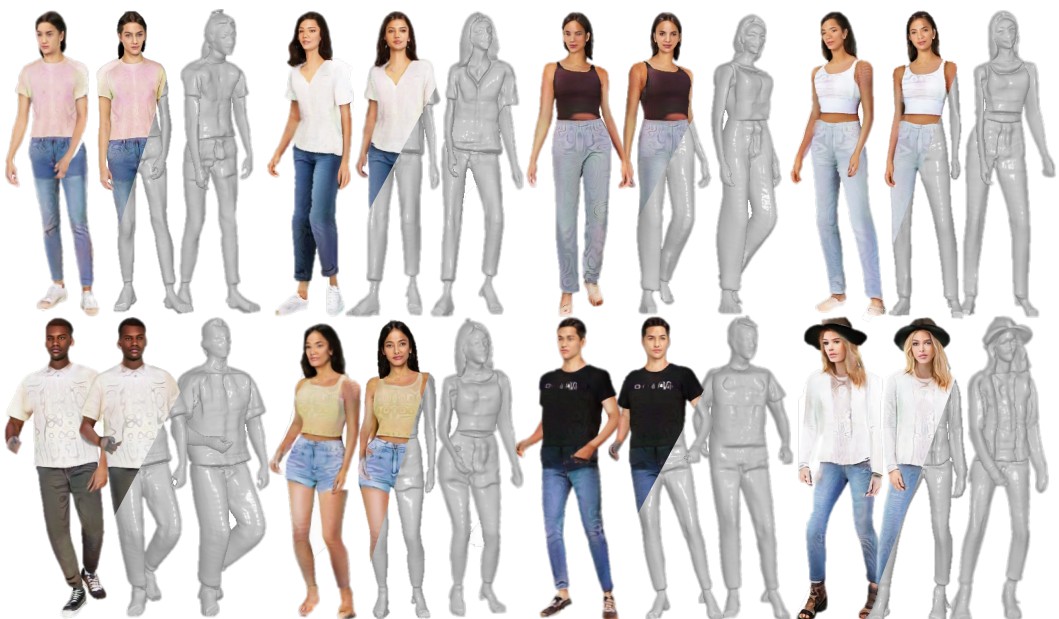

Figure 1: EVA3D generates high-quality and diverse 3D humans with photo-realistic RGB renderings and detailed geometry. Only 2D image collections are used for training.

## ABSTRACT

Inverse graphics aims to recover 3D models from 2D observations. Utilizing differentiable rendering, recent 3D-aware generative models have shown impressive results of rigid object generation using 2D images. However, it remains challenging to generate articulated objects, like human bodies, due to their complexity and diversity in poses and appearances. In this work, we propose, **EVA3D**, an unconditional 3D human generative model learned from 2D image collections only. EVA3D can sample 3D humans with detailed geometry and render high-quality images (up to $512 \times 256$) without bells and whistles (*e.g.* super resolution). At the core of EVA3D is a compositional human NeRF representation, which divides the human body into local parts. Each part is represented by an individual volume. This compositional representation enables **1)** inherent human priors, **2)** adaptive allocation of network parameters, **3)** efficient training and rendering. Moreover, to accommodate for the characteristics of sparse 2D human image collections (*e.g.* imbalanced pose distribution), we propose a pose-guided sampling strategy for better GAN learning. Extensive experiments validate that EVA3D achieves state-of-the-art 3D human generation performance regarding both geometry and texture quality. Notably, EVA3D demonstrates great potential and scalability to "inverse-graphics" diverse human bodies with a clean framework. Our code is publicly available at https://github.com/hongfz16/EVA3D.

## 1   INTRODUCTION

Inverse graphics studies inverse-engineering of projection physics, which aims to recover the 3D world from 2D observations. It is not only a long-standing scientific quest, but also enables nu-

merous applications in VR/AR and VFX. Recently, 3D-aware generative models (Chan et al., 2021; Or-El et al., 2022; Chan et al., 2022) demonstrate great potential in inverse graphics by learning to generate 3D rigid objects (*e.g.* human/animal faces, CAD models) from 2D image collections. However, human bodies, as articulated objects, have complex articulations and diverse appearances. Therefore, it is challenging to learn 3D human generative models that can synthesis animatable 3D humans with high-fidelity textures and vivid geometric details.

To generate high-quality 3D humans, we argue that two main factors should be properly addressed: **1) 3D human representation**; **2) generative network training strategies**. Due to the articulated nature of human bodies, a desirable human representation should be able to explicitly control the pose/shape of 3D humans. With an articulated representation, a 3D human is modeled in its canonical pose (canonical space), and can be rendered in different poses and shapes (observation space). Moreover, the efficiency of the representation matters in high-quality 3D human generation. Previous methods (Noguchi et al., 2022; Bergman et al., 2022) fail to achieve high resolution generation due to their inefficient human representations.

In addition, training strategies could also highly influence 3D human generative models. The issue mainly comes from the data characteristics. Compared with datasets used by Noguchi et al. (2022) (*e.g.* AIST (Tsuchida et al., 2019)), fashion datasets (*e.g.* DeepFashion (Liu et al., 2016)) are more aligned with real-world human image distributions, making a favorable dataset choice. However, fashion datasets mostly have **very limited human poses** and **highly imbalanced viewing angles**. This imbalanced 2D data distribution could hinder 3D GAN learning, leading to difficulties in novel view/ pose synthesis. Therefore, a proper training strategy is in need to alleviate the issue.

In this work, we propose **EVA3D**, an unconditional high-quality 3D human generative model from sparse 2D human image collections only. To facilitate that, we propose a compositional human NeRF representation to improve the model efficiency. We divide the human body into 16 parts and assign each part an individual network, which models the corresponding local volume. Our human representation mainly provides three advantages. **1)** It inherently describes the human body prior, which supports explicit control over human body shapes and poses. **2)** It supports adaptively allocating computation resources. More complex body parts (*e.g.* heads) can be allocated with more parameters. **3)** The compositional representation enables efficient rendering and achieves high-resolution generation. Rather than using one big volume (Bergman et al., 2022), our compositional representation tightly models each body part and prevents wasting parameters on empty volumes. Moreover, thanks to the part-based modeling, we can efficiently sample rays inside local volumes and avoid sampling empty spaces. With the compact representation together with the efficient rendering algorithm, we achieve high-resolution ($512 \times 256$) rendering and GAN training without using super-resolution modules, while existing methods can only train at a native resolution of $128^2$. Moreover, we carefully design training strategies to address the human pose and viewing angle imbalance issue. We analyze the head-facing angle distribution and propose a pose-guided sampling strategy to help effective 3D human geometry learning.

Quantitative and qualitative experiments are performed on two fashion datasets (Liu et al., 2016; Fu et al., 2022) to demonstrate the advantages of EVA3D. We also experiment on UBCFashion (Zablotskaia et al., 2019) and AIST (Tsuchida et al., 2019) for comparison with prior work. Extensive experiments on our method designs are provided for further analysis. In conclusion, our contributions are as follows: **1)** We are the first to achieve high-resolution high-quality 3D human generation from 2D image collections; **2)** We propose a compositional human NeRF representation tailored for efficient GAN training; **3)** Practical training strategies are introduced to address the imbalance issue of real 2D human image collections. **4)** We demonstrate applications of EVA3D, *i.e.* interpolation and GAN inversion, which pave way for further exploration in 3D human GAN.

## 2 RELATED WORK

**3D-Aware GAN.** Generative Adversarial Network (GAN) (Goodfellow et al., 2020) has been a great success in 2D image generation (Karras et al., 2019; 2020). Many efforts have also been put on 3D-aware generation. Nguyen-Phuoc et al. (2019); Henzler et al. (2019) use voxels, and Pan et al. (2020) use meshes to assist the 3D-aware generation. With recent advances in NeRF (Mildenhall et al., 2020; Tewari et al., 2021), many have build 3D-aware GANs based on NeRF (Schwarz et al., 2020; Niemeyer & Geiger, 2021; Chan et al., 2021; Deng et al., 2022). To increase the generation

resolution, Gu et al. (2021); Or-El et al. (2022); Chan et al. (2022) use 2D decoders for super resolution. Moreover, it is desirable to lift the raw resolution, by improving the rendering efficiency, for more detailed geometry and better 3D consistency (Skorokhodov et al., 2022; Xiang et al., 2022; Schwarz et al., 2022; Zhao et al., 2022). We also propose an efficient 3D human representation to allow high resolution training.

**Human Generation.** Though great success has been achieved in generating human faces, it is still challenging to generate human images for the complexity in human poses and appearances (Sarkar et al., 2021b; Lewis et al., 2021; Sarkar et al., 2021a; Jiang et al., 2022c). Recently, Fu et al. (2022); Frühstück et al. (2022) scale-up the dataset and achieve impressive 2D human generation results. For 3D human generation, Chen et al. (2022) generate human geometry using 3D human dataset. Some also attempt to train 3D human GANs using only 2D human image collections. Grigorev et al. (2021); Zhang et al. (2021) use CNN-based neural renderers, which cannot guarantee 3D consistency. Noguchi et al. (2022) use human NeRF (Noguchi et al., 2021) for this task, which only trains at low resolution. Bergman et al. (2022); Zhang et al. (2022a) propose to increase the resolution by super-resolution, which still fails to produce high-quality results. Hong et al. (2022b); Zhang et al. (2022b) generate 3D avatars and motions from text inputs.

**3D Human Representations.** 3D human representations serve as fundamental tools for human related tasks. Loper et al. (2015); Pavlakos et al. (2019b) create parametric human models, for explicit modeling of 3D humans. To model human appearances, Habermann et al. (2021); Shysheya et al. (2019); Yoon et al. (2021); Liu et al. (2021) further introduce UV maps. Parametric modeling gives robust control over the human model, but less realism. Palafox et al. (2021) use implicit functions to generate realistic 3D human body shapes. Embracing the development of NeRF, the number of works about human NeRF has also exploded (Peng et al., 2021b; Zhao et al., 2021; Peng et al., 2021a; Xu et al., 2021; Noguchi et al., 2021; Weng et al., 2022; Chen et al., 2021; Su et al., 2021; Jiang et al., 2022a;b; Wang et al., 2022). Hong et al. (2022a) propose to learn modal-invariant human representations for versatile down-stream tasks. Cai et al. (2022) contribute a large-scale multi-modal 4D human dataset. Some propose to model human body in a compositional way (Mihajlovic et al., 2022; Palafox et al., 2022; Su et al., 2022), where several submodules are used to model different body parts, and are more efficient than single-network ones.

**Compositional NeRF.** The compositional representation has been long studied for its effectiveness and efficiency. It has also been applied to NeRF for object, scene and human modeling. Tancik et al. (2022); Kundu et al. (2022) model outdoor scene NeRF in an compositional way by splitting scenes into block or object levels. Yang et al. (2021); Driess et al. (2022); Wang et al. (2021) decompose multi-objects in a scene for further editing. Compositional NeRF generation has also been studied in prior arts (Niemeyer & Geiger, 2021; BR et al., 2022).

## 3 METHODOLOGY

### 3.1 PREREQUISITES

**NeRF** (Mildenhall et al., 2020) is an implicit 3D representation, which is capable of photorealistic novel view synthesis. NeRF is defined as $\{c, \sigma\} = F_\Phi(x, d)$, where $x$ is the query point, $d$ is the viewing direction, $c$ is the emitted radiance (RGB value), $\sigma$ is the volume density. To get the RGB value $C(r)$ of some ray $r(t) = o + td$, namely volume rendering, we have the following formulation, $C(r) = \int_{t_n}^{t_f} T(t)\sigma(r(t))c(r(t), d)dt$, where $T(t) = \exp(-\int_{t_n}^{t} \sigma(r(s))ds)$ is the accumulated transmittance along the ray $r$ from $t_n$ to $t$. $t_n$ and $t_f$ denotes the near and far bounds. To get the estimation of $C(r)$, it is discretized as

$$\hat{C}(r) = \sum_{i=1}^{N} T_i(1 - \exp(-\sigma_i\delta_i))c_i, \text{ where } T_i = \exp(-\sum_{j=1}^{i-1} \sigma_j\delta_j), \delta_i = t_{i+1} - t_i. \quad (1)$$

For better geometry, Or-El et al. (2022) propose to replace the volume density $\sigma(x)$ with SDF values $d(x)$ to explicitly define the surface. SDF can be converted to the volume density as $\sigma(x) = \alpha^{-1}\text{sigmoid}(-d(x)/\alpha)$, where $\alpha$ is a learnable parameter. In later experiments, we mainly use SDF as the implicit geometry representation, which is denoted as $\sigma$ for convenience.

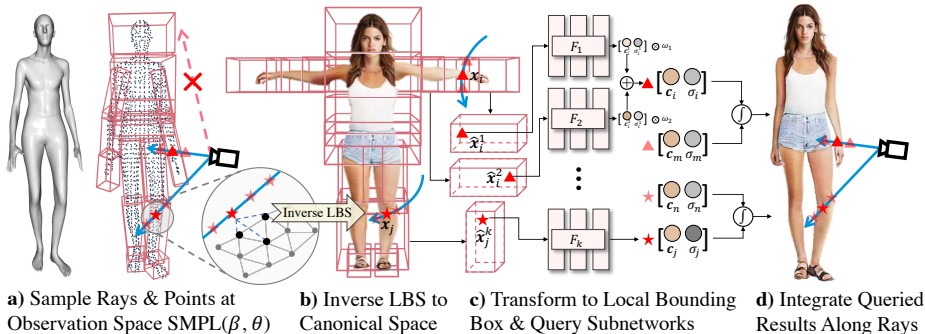

**a)** Sample Rays & Points at Observation Space SMPL($\beta$, $\theta$)    **b)** Inverse LBS to Canonical Space    **c)** Transform to Local Bounding Box & Query Subnetworks    **d)** Integrate Queried Results Along Rays

Figure 2: **Rendering Process of the Compositional Human NeRF Representation.** For shape and pose specified by SMPL($\beta$, $\theta$), local bounding boxes are constructed. Rays that intersect with bounding boxes are sampled and transferred to the canonical space using inverse LBS. Subnetworks corresponding to bounding boxes are queried, results of which are integrated to produce final renderings.

**SMPL** (Loper et al., 2015), defined as $M(\beta, \theta)$, is a parametric human model, where $\beta, \theta$ controls body shapes and poses. In this work, we use the Linear Blend Skinning (LBS) algorithm of SMPL for the transformation from the canonical space to observation spaces. Formally, point $x$ in the canonical space is transformed to an observation space defined by pose $\theta$ as $x' = \sum_{k=1}^{K} w_k G_k(\theta, J) x$, where $K$ is the joint number, $w_k$ is the blend weight of $x$ against joint $k$, $G_k(\theta, J)$ is the transformation matrix of joint $k$. The transformation from observation spaces to the canonical space, namely inverse LBS, takes a similar formulation with inverted transformation matrices.

### 3.2 COMPOSITIONAL HUMAN NERF REPRESENTATION

The compositional human NeRF representation is defined as $\mathbb{F}_\Phi$, corresponding to a set of local bounding boxes $\mathbb{B}$. For each body part $k$, we use a subnetwork $F_k \in \mathbb{F}_\Phi$ to model the local bounding box $\{b_{min}^k, b_{max}^k\} \in \mathbb{B}$, as shown in Fig. 2 b). For some point $x_i$ in the canonical coordinate with direction $d_i$ and falling inside the $k$-th bounding box, the corresponding radiance $c_i^k$ and density $\sigma_i^k$ is queried by

$$\{c_i^k, \sigma_i^k\} = F_k(\hat{x}_i^k, d_i), \text{ where } \hat{x}_i^k = \frac{2x_i - (b_{min}^k + b_{max}^k)}{b_{max}^k - b_{min}^k}. \tag{2}$$

If the point $x_i$ falls in multiple bounding boxes $\mathbb{A}_i$, a window function (Lombardi et al., 2021) is applied to linearly blend queried results. The blended radiance $c_i$ and density $\sigma_i$ of $x_i$ is calculated as

$$\{c_i, \sigma_i\} = \frac{1}{\sum \omega_a} \sum_{a \in \mathbb{A}_i} \omega_a \{c_i^k, \sigma_i^k\}, \text{ where } \omega_a = \exp(-m(\hat{x}_i^k(x)^n + \hat{x}_i^k(y)^n + \hat{x}_i^k(z)^n)). \tag{3}$$

$m, n$ are chosen empirically. Different from Palafox et al. (2022); Su et al. (2022), we only query subnetworks whose bounding boxes contain query points. It increases the efficiency of the query process and saves computational resources.

Taking advantages of the compositional representation, we also adopt an efficient volume rendering algorithm. Previous methods need to sample points, query, and integrate for every pixel of the canvas, which wastes large amounts of computational resources on backgrounds. In contrast, for the compositional representation, we have pre-defined bounding boxes to filter useful rays, which is also the key for our method being able to train on high resolution.

As shown in Fig. 2, for the target pose $\theta$, shape $\beta$ and camera setup, our rendering algorithm $\mathcal{R}(\mathbb{F}_\Phi, \beta, \theta, \text{cam})$ is described as follows. Firstly, ray $r(t) = o + td$ is sampled for each pixel on the canvas. Then we transform the pre-defined bounding boxes $\mathbb{B}$ to the target pose $\theta$ using transformation matrices $G_k$ defined by SMPL. Rays that intersect with the transformed bounding boxes are kept for further rendering. Others are marked to be the background color. For ray $r(t) = o + td$ that intersects with single or multiple bounding boxes, we get the near and far bounds $t_n, t_f$.

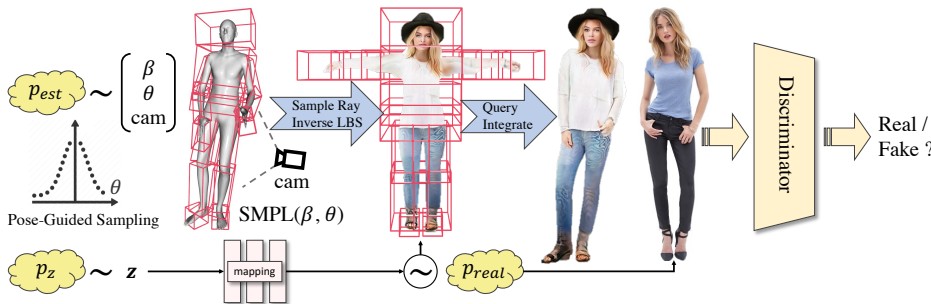

Figure 3: **3D Human GAN Framework.** With the estimated SMPL and camera parameters distribution $p_{est}$, 3D humans are randomly sampled and rendered conditioned on $z \sim p_z$. The renderings are used for adversarial training against real 2D human image collections $p_{real}$.

$N$ points are randomly sampled on each ray as

$$t_i \sim \mathcal{U}\left[t_n + \frac{i-1}{N}(t_f - t_n), t_n + \frac{i}{N}(t_f - t_n)\right]. \tag{4}$$

Next, we transform sampled points back to the canonical space using inverse LBS. Similar to Zheng et al. (2021), we inverse not only the pose transformation, but also the shape/ pose blend shapes $\boldsymbol{B_S}(\boldsymbol{\beta}), \boldsymbol{B_P}(\boldsymbol{\theta})$ to be able to generalize to different body shapes. For sampled point $\boldsymbol{r}(t_i)$, the nearest $k$ points $\mathbb{N} = \{\boldsymbol{v}_1 ... \boldsymbol{v}_k\}$ are found among the vertices of the posed SMPL mesh $M(\boldsymbol{\beta}, \boldsymbol{\theta})$. The transformation of point $\boldsymbol{r}(t_i)$ from the observation space to the canonical space is defined as

$$\begin{bmatrix} \boldsymbol{x}_i^0 \\ \mathbf{1} \end{bmatrix} = \sum_{v_j \in \mathbb{N}} \frac{\omega_j}{\sum \omega_j}(\boldsymbol{M}_j)^{-1} \begin{bmatrix} \boldsymbol{r}(t_i) \\ \mathbf{1} \end{bmatrix}, \text{ where } \boldsymbol{M}_j = \left(\sum_{k=1}^{K} w_k^j \boldsymbol{G}_k\right) \begin{bmatrix} \boldsymbol{I} & \boldsymbol{B}_S^j + \boldsymbol{B}_P^j \\ \mathbf{0} & \boldsymbol{I} \end{bmatrix}. \tag{5}$$

$\omega_j = 1/\|\boldsymbol{r}(t_i) - \boldsymbol{v_j}\|$ is the inverse distance weight. $\boldsymbol{M}_j$ is the transformation matrix of the SMPL vertex $v_j$. Then we query the compositional human NeRF representation $\mathbb{F}$ with point $\boldsymbol{x}_i^0$ to get its corresponding radiance $\boldsymbol{c}_i$ and density $\sigma_i$ as defined in Eq. 2 and 3. Finally, we integrate the queried results for the RGB value of ray $\boldsymbol{r}(t)$, as defined in Eq. 1.

### 3.3 3D HUMAN GAN FRAMEWORK

With the compositional human NeRF representation, we construct a 3D human GAN framework as shown in Fig. 3. The generator is defined as $G(\boldsymbol{z}, \boldsymbol{\beta}, \boldsymbol{\theta}, \text{cam}; \Phi_G) = \mathcal{R}(\mathbb{F}_{\Phi_G}(\boldsymbol{z}), \boldsymbol{\beta}, \boldsymbol{\theta}, \text{cam})$. Similar to pi-GAN (Chan et al., 2021), each subnetwork of $\mathbb{F}_\Phi$ consists of stacked MLPs with SIREN activation (Sitzmann et al., 2020). To generate fake samples, $\boldsymbol{z} \sim p_{\boldsymbol{z}}$ is sample from normal distribution. $\{\boldsymbol{\beta}, \boldsymbol{\theta}, \text{cam}\} \sim p_{est}$ are sampled from the estimated distribution from 2D image collections. We use off-the-shelf tools (Pavlakos et al., 2019a; Kocabas et al., 2020) to estimate $\{\boldsymbol{\beta}, \boldsymbol{\theta}, \text{cam}\}$ for the 2D image collections. Unlike ENARF-GAN(Noguchi et al., 2022), where these variables are sampled from the distribution of motion datasets (Mahmood et al., 2019), the real 2D image collections do not necessarily share the similar pose distribution as that of motion datasets, especially for fashion datasets, *e.g.* DeepFashion, where the pose distribution is imbalanced. Finally, the fake samples $\boldsymbol{I}_f = G(\boldsymbol{z}, \boldsymbol{\beta}, \boldsymbol{\theta}, \text{cam}; \Phi_G)$, along with real samples $\boldsymbol{I}_r \sim p_{real}$ are sent to discriminator $D(I; \Phi_D)$ for adversarial training. For more implementation details, please refer to the supplementary material.

### 3.4 TRAINING

**Delta SDF Prediction.** Real-world 2D human image collections, especially fashion datasets, usually have imbalanced pose distribution. For example, as shown in Fig. 6, we plot the distribution of facing angles of DeepFashion. Such heavily imbalanced pose distribution makes it hard for the network to learn correct 3D information in an unsupervised way. Therefore, we propose to introduce strong human prior by utilizing the SMPL template geometry $\boldsymbol{d}_T(\boldsymbol{x})$ as the foundation of our human representation. Instead of directly predicting the SDF value $\boldsymbol{d}(\boldsymbol{x})$, we predict an SDF offset $\Delta \boldsymbol{d}(\boldsymbol{x})$ from the template (Yifan et al., 2022). Then $\boldsymbol{d}_T(\boldsymbol{x}) + \Delta \boldsymbol{d}(\boldsymbol{x})$ is used as the actual SDF value of point $\boldsymbol{x}$.

**Pose-guided Sampling.** To facilitate effective 3D information learning from sparse 2D image collections, other than introducing a 3D human template, we propose to balance the input 2D images

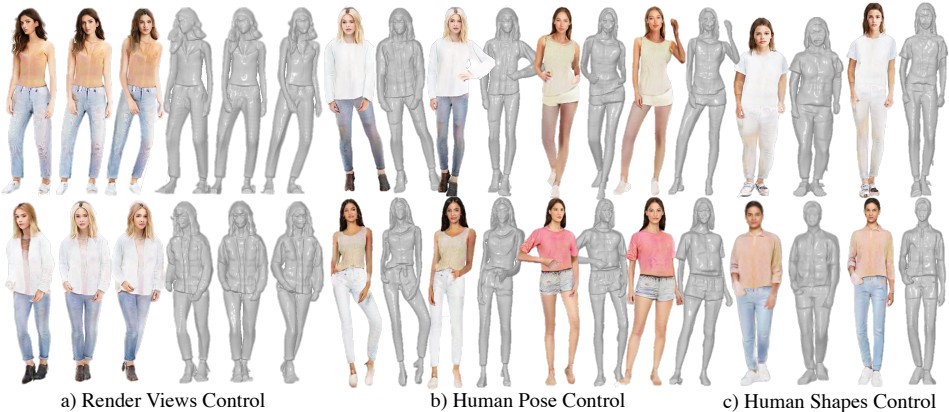

a) Render Views Control        b) Human Pose Control        c) Human Shapes Control

Figure 4: **Generation Results of EVA3D.** The 3D-aware nature and inherent human prior of EVA3D enable explicit control over rendering views, human poses, and shapes.

based on human poses. The intuition behind the pose-guided sampling is that different viewing angles should be sampled more evenly to allow effective learning of geometry. Empirically, among all human joints, we use the angle of the head to guide the sampling. Moreover, facial areas contain more information than other parts of the head. Front-view angles should be sampled more than other angles. Therefore, we choose to use a Gaussian distribution centered at the front-view angle $\mu_\theta$, with a standard deviation of $\sigma_\theta$. Specifically, $M$ bins are divided on the circle. For an image with the head angle falling in bin $m$, its probability $p_m$ of being sampled is defined as

$$p_m = \frac{1}{\sigma_\theta \sqrt{2\pi}} \exp\left(-\frac{1}{2}\left(\frac{\theta_m - \mu_\theta}{\sigma_\theta}\right)^2\right), \text{ where } \theta_m = \frac{2\pi m}{M}. \tag{6}$$

We visualize the balanced distribution in Fig. 6. The network now has higher chances of seeing more side-views of human bodies, which helps better geometry generation.

**Loss Functions.** For the adversarial training, we use the non-saturating GAN loss with R1 regularization (Mescheder et al., 2018), which is defined as

$$\mathcal{L}_{\text{adv}}(\Phi_G, \Phi_D) = \boldsymbol{E}_{\boldsymbol{z} \sim p_z, \{\boldsymbol{\beta}, \boldsymbol{\theta}, \text{cam}\} \sim p_{est}}[f(D(G(\boldsymbol{z}, \boldsymbol{\beta}, \boldsymbol{\theta}, \text{cam}; \Phi_G); \Phi_D))] \tag{7}$$
$$+ \boldsymbol{E}_{\boldsymbol{I}_r \sim p_{real}}[f(-D(\boldsymbol{I}_r; \Phi_D)) + \lambda|\nabla D(\boldsymbol{I}_r; \Phi_D)|^2], \tag{8}$$

where $f(u) = -\log(1 + \exp(-u))$. Other than the adversarial loss, some regularization terms are introduced for the delta SDF prediction. Firstly, we want minimum offset from the template mesh to maintain plausible human shape, which gives the minimum offset loss $\mathcal{L}_{\text{off}} = \boldsymbol{E}_{\boldsymbol{x}}[\|\Delta d(\boldsymbol{x})\|_2^2]$. Secondly, to ensure that the predicted SDF values are physically valid (Gropp et al., 2020), we penalize the derivation of delta SDF predictions to zero $\mathcal{L}_{\text{eik}} = \boldsymbol{E}_{\boldsymbol{x}}[\|\nabla(\Delta d(\boldsymbol{x}))\|_2^2]$. The overall loss is defined as $\mathcal{L} = \mathcal{L}_{\text{adv}} + \lambda_{\text{off}}\mathcal{L}_{\text{off}} + \lambda_{\text{eik}}\mathcal{L}_{\text{eik}}$, where $\lambda_*$ are loss weights defined empirically.

## 4 EXPERIMENTS

### 4.1 EXPERIMENTAL SETUP

**Datasets.** We conduct experiments on four datasets: DeepFashion (Liu et al., 2016), SHHQ (Fu et al., 2022), UBCFashion (Zablotskaia et al., 2019) and AIST (Tsuchida et al., 2019). The first two are sparse 2D image collections, meaning that each image has different identities and poses are sparse, which makes them more challenging. The last two are human video datasets containing different poses/ views of the same identities, which is easier for the task but lacks diversity.

**Comparison Methods.** We compare with three baselines. ENARF-GAN (Noguchi et al., 2022) makes the first attempt at human NeRF generation from 2D image collections. EG3D (Chan et al., 2022) and StyleSDF (Or-El et al., 2022) are state-of-the-art methods for 3D-aware generation, both requiring super-resolution modules to achieve high-resolution generation.

**Evaluation Metrics.** To evaluate the quality of rendered images, we adopt Frechet Inception Distance (FID) (Heusel et al., 2017) and Kernel Inception Distance (KID) (Bińkowski et al., 2018).

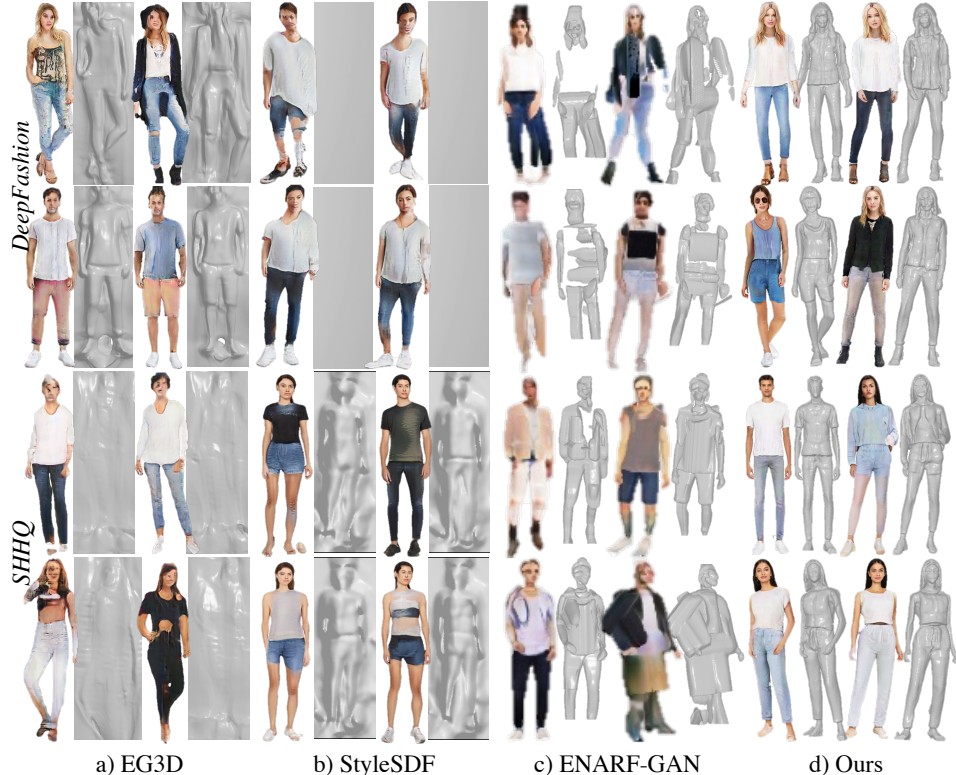

a) EG3D        b) StyleSDF        c) ENARF-GAN        d) Ours

Figure 5: **Qualitative Comparison Between EVA3D and Baseline Methods.** Rendered 2D images and corresponding meshes are placed side-by-side. Both the 2D renderings and 3D meshes generated by our method achieve the best quality among SOTA methods. Zoom in for the best view.

Following ENARF-GAN, we use Percentage of Correct Keypoints (PCKh@0.5) (Andriluka et al., 2014) to evaluate the correctness of generated poses. Note that PCKh@0.5 can only be calculated on methods that can control generated poses, *i.e.* ENARF-GAN and EVA3D. To evaluate the correctness of geometry, we use an off-the-shelf tool (Ranftl et al., 2022) to estimate depth from the generated images and compare it with generated depths. $50K$ samples, padded to square and resized to the same resolution (DeepFashion, SHHQ, UBCFashion at $512^2$; AIST at $256^2$), are used to compute FID and KID. PCKh@0.5 and Depth are evaluated on $5K$ samples.

## 4.2 QUALITATIVE EVALUATIONS

**Generation Results and Controlling Ability of EVA3D.** As shown in Fig. 4 a), EVA3D is capable of generating high-quality renderings in novel views and remain multi-view consistency. Due to the inherent human prior in our model design, EVA3D can control poses and shapes of the generated 3D human by changing $\beta$ and $\theta$ of SMPL. We show novel pose and shape generation results in Fig. 4 b)& c). We refer readers to the supplementary PDF and video for more qualitative results.

**Comparison with Baseline Methods.** We show the renderings and corresponding meshes generated by baselines and our method in Fig. 5. EG3D trained on DeepFashion, as well as StyleSDF trained on SHHQ, generate reasonable RGB renderings and geometry. However, without explicit human modeling, complex human poses make it hard to align and model 3D humans in observation spaces, which leads to distorted generation. Moreover, because of the use of super resolution, their geometry is only trained under low resolution ($64^2$) and therefore lacks details. EG3D trained on SHHQ and StyleSDF trained on DeepFashion fail to capture 3D information and collapse to the trivial solution of painting on billboards. Limited by the inefficient representation and computational resources, ENARF-GAN can only be trained at a resolution of $128^2$, which leads to low-quality rendering results. Besides, lacking human prior makes ENARF-GAN hard to capture correct 3D information of human from sparse 2D image collections, which results in broken meshes. EVA3D, in contrast, generates high-quality human renderings on both datasets. We also succeeded in learning reasonable 3D human geometry from 2D image collections with sparse viewing angles and poses, thanks to the strong human prior and the pose-guided sampling strategy. Due to space limitations, we only show

Table 1: Comparison with State-of-the-Art Methods. * The training code of ENARF-GAN is implemented based on the official inference code.

| Methods, Resolution | *DeepFashion* | | | | *SHHQ* | | | |
|---|---|---|---|---|---|---|---|---|
| | FID↓ | KID↓ | PCK↑ | Depth↓ | FID↓ | KID↓ | PCK↑ | Depth↓ |
| EG3D, $512^2$ | 26.38 | 0.014 | - | 0.0779 | 32.96 | 0.033 | - | 0.0296 |
| StyleSDF, $512^2$ | 92.40 | 0.136 | - | 0.0359 | 14.12 | 0.010 | - | 0.0300 |
| ENARF-GAN*, $128^2$ | 77.03 | 0.114 | 43.74 | 0.1151 | 80.54 | 0.102 | 40.17 | 0.1241 |
| Ours, $512^2$ | **15.91** | **0.011** | **87.50** | **0.0272** | **11.99** | **0.009** | **88.95** | **0.0177** |
| Methods, Resolution | *UBCFashion* | | | | *AIST* | | | |
| | FID↓ | KID↓ | PCK↑ | Depth↓ | FID↓ | KID↓ | PCK↑ | Depth↓ |
| EG3D, $512^2$ | 23.95 | **0.009** | - | 0.1163 | 34.76 | 0.022 | - | 0.1165 |
| StyleSDF, $512^2$ | 18.52 | 0.011 | - | 0.0311 | 199.5 | 0.225 | - | 0.0236 |
| ENARF-GAN*, $128^2$ | - | - | - | - | 73.07 | 0.075 | 42.85 | 0.1128 |
| Ours, $512^2$ | **12.61** | 0.010 | **99.17** | **0.0090** | **19.40** | **0.010** | **83.15** | **0.0126** |

Table 2: Ablation Study. [†]Depth is evaluated with SMPL depth. We report Depth$\times 10^3$ for simplicity.

| Methods | FID↓ | [†]Depth↓ |
|---|---|---|
| Baseline, $256^2$ | 31.14 | 3.57 |
| + Composite, $512^2$ | 17.81 | 5.02 |
| + Delta SDF, $512^2$ | **15.62** | 3.69 |
| + Pose-guide, $512^2$ | 15.91 | **3.04** |

Table 3: Trade-Off Between RGB and Geometry.

| Distribution | FID↓ | [†]Depth↓ |
|---|---|---|
| Original | 15.62 | 3.69 |
| $\sigma_\theta = 15°$ | 15.91 | 3.04 |
| $\sigma_\theta = 30°$ | 19.05 | 2.58 |
| $\sigma_\theta = 45°$ | 19.56 | 2.65 |
| $\sigma_\theta = 60°$ | 25.08 | 2.91 |
| Uniform | 25.82 | 2.92 |

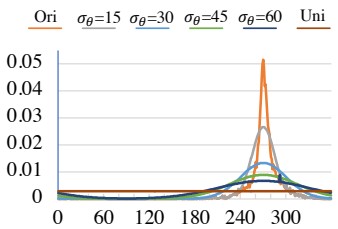

Figure 6: PDF of Different Pose-Guided Sampling Distributions.

results of DeepFashion and SHHQ here. For visual comparisons on UBCFashion and AIST, please refer to the supplementary material.

### 4.3 QUANTITATIVE EVALUATIONS

As shown in Tab. 1, our method leads all metrics in four datasets. EVA3D outperforms ENARF-GAN in all settings thanks to our high-resolution training ability. EG3D and StyleSDF, as the SOTA methods in the 3D generation, can achieve reasonable scores in some settings (*e.g.* StyleSDF achieves 18.52 FID on UBCFashion) for their super-resolution modules. But they also fail on some datasets (*e.g.* StyleSDF fails on AIST with 199.5 FID) for complexity in human poses. In the contrast, EVA3D achieves the best FID/KID scores under all settings. Moreover, unlike EG3D or StyleSDF, EVA3D can control the generated pose and achieve higher PCKh@0.5 score than ENARF-GAN. For the geometry part, we also achieve the lowest depth error, which shows the importance of natively high-resolution training.

### 4.4 ABLATION STUDIES

**Ablation on Method Designs.** To validate the effectiveness of our designs on EVA3D, we subsequently add different designs on a baseline method, which uses one large network to model the canonical space. Experiments are conducted on DeepFashion. The results are reported in Tab. 2. Limited by the inefficient representation, the "Baseline" can only be trained at $256 \times 128$, which results in the worst FID score. Adding compositional design ("+Composite") makes the network efficient enough to be trained at a higher resolution of $512 \times 256$ and achieve higher generation quality. We further introduce human prior by predicting delta SDF ("+Delta SDF"), which gives the best FID score. Finally, using the pose-guided sampling ("+Pose-guide"), we further decrease the depth error. We refer readers to the supplementary material for qualitative evaluations of ablation studies.

**Analysis on Pose-Guided Sampling.** We analyze the importance of the sampling strategy in 3D human GAN training. Three types of distributions $p_{est}$ are experimented, including the original dataset distribution ("Original"), pose-guided Gaussian distribution ("$\sigma_\theta = *$"), and pose-guided uniform distribution ("Uniform"). The results are reported in Tab. 3. Firstly, uniform sampling is not a good strategy for that the information density is different between different parts of human.

a) Interpolation on Latent Space      b) Target     Inversion Result

Figure 7: **Applications of EVA3D.** a) Interpolation on the latent space gives smooth transition between two samples. b) Inversion result (right) of the target image (left).

Secondly, the original distribution gives the best visual quality but the worst geometry. It could result in the trivial solution of painting on billboards. Thirdly, the pose-guided Gaussian sampling can avoid damaging visual quality too much and improve geometry learning. As the standard deviation $\sigma_\theta$ increases, FID increases while the depth error decreases. Therefore, it is a trade-off between visual quality and geometry quality. In our final experiments, we choose $\sigma_\theta = 15°$ which is a satisfying balance between the two factors.

### 4.5 APPLICATIONS

**Interpolation on Latent Space.** As shown in Fig. 7 a), we linearly interpolate two latent codes to generate a smooth transition between them, showing that the latent space learned by EVA3D is semantically meaningful. More results are provided in the supplementary video.

**Inversion.** We use Pivotal Tuning Inversion (PTI) (Roich et al., 2021) to inverse the target image and show the results in Fig. 7 b). Reasonable novel view synthesis results can be achieved. The geometry, however, fails to capture geometry details corresponding to RGB renderings, which can be caused by the second stage generator fine-tuning of PTI. Nevertheless, we demonstrate the potential of EVA3D in more related downstream tasks. For more results and comparison with baseline methods, please refer to the supplementary material.

## 5 DISCUSSION

To conclude, we propose a high-quality unconditional 3D human generation model EVA3D that only requires 2D image collections for training. We design a compositional human NeRF representation for efficient GAN training. To train on the challenging 2D image collections with sparse viewing angles and human poses, *e.g.* DeepFashion, strong human prior and pose-guided sampling are introduced for better GAN learning. On four large-scale 2D human datasets, we achieve state-of-the-art generation results at a high resolution of $512 \times 256$.

**Limitations: 1)** There still exists visible circular artifacts in the renderings, which might be caused by the SIREN activation. A better base representation, *e.g.* tri-plane of EG3D, and a 2D decoder might solve the issue. **2)** The estimation of SMPL parameters from 2D image collections is not accurate, which leads to a distribution shift from the real pose distribution and possibly compromises generation results. Refining SMPL estimation during training would make a good future work. **3)** Limited by our tight 3D human representation, it is hard to model loose garments, accessories or body parts (like hair). The apparent geometric line artifact around the neck and shoulder areas of samples from DeepFashion training (see Fig. 5) could be caused by the compositional representation having trouble modeling long hair hanging down. Using separate modules to handle loose parts might be a promising direction. **4)** It is known that state-of-the-art 3D-aware generation methods (Chan et al., 2022; Or-El et al., 2022) have not achieved comparable quality with that of 2D generation (Karras et al., 2020; 2021). To investigate if that is still the case in terms of human generation, we further train StyleGAN2 (Karras et al., 2020) on DeepFashion. StyleGAN2 achieves **6.52** FID, which is much lower than our FID of **15.91**. This indicates that 3D-aware generation still has a long way to develop.

### ACKNOWLEDGMENTS

This study is supported by NTU NAP, MOE AcRF Tier 2 (MOE-T2EP20221-0012), and under the RIE2020 Industry Alignment Fund – Industry Collaboration Projects (IAF-ICP) Funding Initiative, as well as cash and in-kind contribution from the industry partner(s).

ETHICS STATEMENT

Although the results of EVA3D are yet to the point where they can fake human eyes, we still need to be aware of its potential ethical issues. The generated 3D humans might be misused to create contents that are misleading. EVA3D can also be used to invert real human images, which can be used to create fake videos of real humans and cause negative social impacts. Moreover, the generated 3D humans might be biased, which is caused by the inherent distribution of training datasets. We make our best effort to demonstrate the impartiality of EVA3D in Fig. 1.

REPRODUCIBILITY STATEMENT

Our method is thoroughly described in Sec. 3. Together with implementation details included in the supplementary material, the reproducibility is ensured. Moreover, Our code is publicly available at `https://github.com/hongfz16/EVA3D`.

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
