# OpenReview forum: "EVA3D: Compositional 3D Human Generation from 2D Image Collections"
_ICLR.cc/2023/Conference — ICLR 2023 notable top 25%_

### Official Review · Reviewer_bCYZ · 2022-10-22

**Confidence:** 3
**Correctness:** 3
**Technical Novelty And Significance:** 3
**Empirical Novelty And Significance:** 3
**Recommendation:** 8

**Clarity, Quality, Novelty And Reproducibility:**

The paper is in general well-written and easy to follow. With the details provided in the paper, one should be able to implement with sufficient tuning. It might be hard to reproduce the exact results but the author promised to release the code.

**Strength And Weaknesses:**

Strength:
1.  the overall pipeline is well-designed and with promising performances.  The involving of SMPL as human  human priors and porposed pose-guided sampling strategy are adaptive and efficient.

2.  given the evaluation metrics, the proposed method outperforms SOTA of 3D body generative model by a large margin.

3.  detailed ablation for the method design and pose sampling.

Weaknesses:
1.  technical side, the novelty is kind limited, as it is not totally new to use a multi-nerf based methods to represent objects/scenes (it would also be better to add discussion about early works like BlockNerf in the related work).

2. the results of the interpolation and inversion are interesting but no baseline was provided and the results are a bit blur.



**Summary Of The Paper:**

This paper presents a 3D human generative model which can learn to generate full body from 2D image collections. It proposes to decompose the human body into 16 parts and represent each part with a separate network. It also adapts several strategies for efficient training and sampling.  It also provides  evaluations and shows improvements over  pervious SOTAs in both generated geometry and texture quality.



**Summary Of The Review:**

Overall, I think the proposed method is sound with promising performance. Although it is not new to represent with multi-nerf, the method is well-designed and with efficient adaption. Hence I am in favor for acceptance.

---

> ### Author Response · Authors · 2022-11-14
> **Response to Reviewer bCYZ**
>
> Thank you for your careful review and constructive suggestions. We summarise and answer your questions below.
>
> > **Q1: It is not new to use multi-nerf based method to represent objects/scenes**
>
> Please kindly refer to the above post named "Response to the Common Issue of the Novelty of the Compositional Representation".
>
> > **Q2: No baseline was provided for interpolation and inversion**
>
> Thank you for the suggestion. To show our advantages on latent code interpolation and inversion over baseline methods, we further add visual comparisons in our revision (please kindly refer to supplementary material Fig.5). Because StyleSDF has no control over human poses, pose changes are observed (see left hands) when interpolating between latent codes. The texture stitching problem, as pointed out in StyleGAN3, can also be seen during pose changing. Moreover, some middle results are semantically incorrect due to inferior generation quality of StyleSDF. In contrast, EVA3D demonstrates consistently high-quality interpolation results during latent space interpolation. For the inversion, StyleSDF fails to recover detailed appearances of input target even with the second stage fine-tuning of PTI. Though the geometry is hard to be inverted with a single image as input, EVA3D manages to recover fine details of the input image. Admittedly, the inverted results are blurry, especially when changing camera views. We think that is because the ambiguity of a single image as input and errors introduced by single image SMPL estimation. Besides, PTI is an inversion method designed for 2D GANs. It might not be suitable for 3D-aware GAN inversion. We expect more explorations in 3D-aware GAN inversion in the future, which might inspire tasks like single image 3D reconstruction and 3D-aware editing. Related discussion is also added in our revision (see supplementary material Section 4.1)
>
> Please do not hesitate to let us know if there are any further clarifications that we can offer.

---

### Official Review · Reviewer_zbyZ · 2022-10-24

**Confidence:** 4
**Correctness:** 4
**Technical Novelty And Significance:** 3
**Empirical Novelty And Significance:** 3
**Recommendation:** 6

**Clarity, Quality, Novelty And Reproducibility:**

- It is good to clarify and note in the paper how FID is computed in Table 2, where the resolution of the baseline and remaining methods differ. It would be fair to compute the FID with a single resolution, e.g., the baseline results should be upsampled.


**Strength And Weaknesses:**

** Strength **

- The proposed method contributes toward improving 3D-aware generative human modeling, which is an active research direction.

- The experiments are compelling for both qualitative and quantitative results.


** Weakness **

1) The compositional representation is a minor improvement compared to existing techniques in NeRF and volume rendering. Applying it for human modeling is a good application, though.

2) It is known that FID/KID of 3D-aware generative models is worse compared to 2D GAN models. It would be good to demonstrate whether this is still the case for human generation.


**Summary Of The Paper:**

The paper presents a method to learn 3D human from 2D image collections. By leveraging a 3D-aware generative model, the authors propose to integrate a compositional representation and a prior by SMPL, and an improved training strategy to enable digital human generation. The authors show that it is possible to sample high-resolution depth and color images of 3D human in a much better quality compared to the previous works.

**Summary Of The Review:**

- The paper demonstrates a good application of 3D-aware generative model for human generation, with the additional tweaks such as the compositional representation, SMPL prior, and pose-guided sampling to enable high-quality results. At the moment I am positive despite that I still have some minor concerns that would be good to address or discuss in the rebuttal period.

---

> ### Author Response · Authors · 2022-11-14
> **Response to Reviewer zbyZ**
>
> Thank you for your careful review and constructive suggestions. We summarize and answer your questions below.
>
> > **Q1: The compositional representation is a minor improvement**
>
> Please kindly refer to the above post named "Response to the Common Issue of the Novelty of the Compositional Representation".
>
> > **Q2: Whether 3D-aware human generation is worse than pure 2D generation**
>
> Thank you for the question. We train StyleGAN2 on DeepFashion and get an FID of **6.52**, while EVA3D achieves an FID of **15.91**. Therefore, 2D GAN still has advantages over 3D-aware GANs in terms of human generation quality. Improving 3D-aware generation towards the quality of state-of-the-art 2D counterparts is definitely one of the most exciting future directions. We also include the results and related discussion in our revision (see main paper Section 5 Limitations 4).
>
> > **Q3: How FID is computed in Tab.2**
>
> Thank you for the question. Apologies for not making it clear in our initial submission. For all experiments (except AIST), we resize generated samples to the same resolution of 512^2 before calculating FID/KID. For AIST, we resize images to 256^2 before calculating metrics. We add this explanation to subsection 4.1 "Evaluation Metrics" in our revision.
>
> Please do not hesitate to let us know if there are any further clarifications that we can offer.

---

### Official Review · Reviewer_eUx6 · 2022-10-25

**Confidence:** 4
**Correctness:** 3
**Technical Novelty And Significance:** 3
**Empirical Novelty And Significance:** 3
**Recommendation:** 6

**Clarity, Quality, Novelty And Reproducibility:**

- Clarify: the paper is clearly written and easy to follow.
- Quality: the results are of high quality.
- Novelty: it is novel to integrate techniques mentioned in the paper into a generative model training pipeline.
- Reproducibility: authors state that the code will be public.

**Strength And Weaknesses:**

## Strengths

The proposed approach is interesting and effective. Even though some techniques are not new, e.g., LBS mapping is commonly used in human reconstruction, to integrate them into generative model training is novel.

## Weakness

I do not find major issues in the paper. However, there are some questions I wish authors can clarify or add to make this paper stronger. See below.

## Questions

**1. Clarification about "Baseline" in Tab. 2**

Can authors clarify the difference between the "Baseline" in Tab. 2 and StyleSDF in Tab. 1? I am a little bit confused because the performance differs quite a lot. My understanding is that Tab. 2's "Baseline" includes the mapping with LBS. Is this correct?

**2. For the effectiveness of composition**

In Tab. 2, to demonstrate the efficacy of composition, authors show the results between $512^2$ and $256^2$. This comparison is great to show the power of composition for high resolution. However, I feel like it may be a more direct comparison if authors can show the results from just $256^2$ with composition.

**3. For the issues arising from composition**

From qualitative results, e.g., Fig. 4 in the supplementary, it seems like the composition can impose apparent artifacts around the edge of the bounding box. For example, there are apparent lines around the neck. It would be great if authors can provide more discussion and analysis of this.

**4. For the training strategies**

The proposed training strategies, i.e., pose-guided training and LBS mapping from observation space to canonical space, are essentially generalizable to all baseline methods (StyleSDF and EG3D). Meanwhile, the Delta prediction can also be applied to StyleSDF. It would build a stronger paper if authors can utilize these training strategies in baselines's training.

This is somehow related to the effectiveness of composition in Question 2: the above experiments would give clearer view of how the composition works compared to baselines's representations.

**5. Lacked references**

- Schwarz et al., VoxGRAF: Fast 3D-Aware Image Synthesis with Sparse Voxel Grids. NeurIPS 2022.
- Zhao et al., Generative multiplane images: making a 2D GAN 3D-aware. ECCV 2022.


**Summary Of The Paper:**

This paper tackles the problem of 3D-aware human generation from 2D images. To generate at a high resolution, EVA3D proposes to
use compositional multiple NeRFs to do the generation. To overcome difficulties in training with imbalanced human dataset, the paper proposes several training strategies. Results on several human dataset demonstrate the effectiveness of the proposed method.

**Summary Of The Review:**

The proposed approach is interesting. The paper is clearly written. Meanwhile, effectiveness of some techniques can be further discussed.

---

> ### Author Response · Authors · 2022-11-14
> **Response to Reviewer eUx6**
>
> Thank you for your careful review and constructive suggestions. We summarize and answer your questions below.
>
> > **Q1: Clarification about "Baseline" in Tab.2**
>
> Yes, your understanding is correct. "Baseline" in Tab.2 models canonical space and uses LBS mapping between canonical and observation space. While StyleSDF in Tab.1 directly models the observation space.
>
> > **Q2: Results of 256^2 with composition**
>
> Thank you for the constructive suggestions. We compute FID score for 256^2 with composition, which is **27.96** and better than the "Baseline" **31.14** FID. It demonstrates that the compositional design improves generation quality under the same resolution.
>
> > **Q3: Issues arising from composition**
>
> Thank you for pointing this out. After careful comparison, we find that line artifacts can only be found on the samples generated from DeepFashion training and can only be found on the necks and shoulders. Moreover, as shown in supplementary Fig.4, the Baseline results do not have line artifacts. Based on the observations, we think the problem should be caused by 1) DeepFashion, 2) the compositional representation and 3) head area modeling. Firstly, DeepFashion only contains 8036 samples, which is an order of magnitude less than other three datasets. It is known that generative models trained with small data are prone to artifacts [1]. Moreover, many models in DeepFashion have long hair hanging in front of the shoulder, causing trouble for the compositional representation to model. The head and shoulder sub-modules try their best to composite the hanging-down hair and manage to give relatively continuous renderings. But due to the lack of 3D supervision, it is difficult for both modules to composite continuous surfaces at edges of their bounding boxes. The bigger limitation behind this observation is that the current compositional representation cannot model loose garments, accessories or other body parts (like hair). Using separate modules to handle loose parts (like hair) might alleviate the problem. We add related discussion in our revision (see main paper Section 5 Limitations 3 and supplementary material Section 4.2).
>
> > **Q4: Training strategies applied to baselines**
>
> Thank you for your kind suggestion. It is true that LBS mapping and delta prediction are generalizable to EG3D and StyleSDF. By applying LBS mapping to StyleSDF, we build Baseline in Tab.2. As discussed above in Q2, the additional experiment validates the effectiveness of the compositional representation. As for delta SDF prediction, it is not trivial to directly apply it in observation space. Since vanilla StyleSDF has no control over generated human poses, it is unclear which pose the template SDF should be in for a random sample.
>
> > **Q5: Lacked references**
>
> Thank you. We have added missing references in our revision (see references in blue texts).
>
> Please do not hesitate to let us know if there are any further clarifications that we can offer.
>
> **References**
>
> [1] Tero Karras, Miika Aittala, Janne Hellsten, Samuli Laine, Jaakko Lehtinen, and Timo Aila. Training generative adversarial networks with limited data. In Proc. NeurIPS, 2020

---

### Author Response · Authors · 2022-11-14
**Response to the Common Issue of the Novelty of the Compositional Representation**

Both Reviewer zbyZ and bCYZ mention that the compositional representation is not totally new and has been used in many existing techniques. Admittedly, the compositional representation, for its efficiency and effectiveness, has been widely used for modeling objects, scenes and humans. Though the idea is not new, it is not trivial to make it work on high-resolution 3D human NeRF generation. Specifically, for previously compositional human representations, e.g. ENARF-GAN [1], SPAMS [2], DANBO [3], one needs to evaluate all networks representing different parts and blend results afterwards to query value for one spatial point, which is computationally inefficient. Instead, we propose to query in a sparse way and only evaluate sub-modules close to the spatial point, which largely saves computational cost and is essential for high-resolution GAN training and generation. Moreover, our work shows that for compositional human representation, it is possible to achieve high-quality rendering without densely querying and blending, which we believe is not only scientifically important but also practical in engineering. As suggested by reviewers, we have also added related discussions in our revision (see main paper Section 2 "Compositional NeRF").

**References**

[1] Atsuhiro Noguchi, Xiao Sun, Stephen Lin, and Tatsuya Harada. Unsupervised learning of efficient geometry-aware neural articulated representations. arXiv preprint arXiv:2204.08839, 2022.

[2] Pablo Palafox, Nikolaos Sarafianos, Tony Tung, and Angela Dai. Spams: Structured implicit parametric models. In Proceedings of the IEEE/CVF Conference on Computer Vision and Pattern Recognition, pp. 12851–12860, 2022.

[3] Shih-Yang Su, Timur Bagautdinov, and Helge Rhodin. Danbo: Disentangled articulated neural body representations via graph neural networks. arXiv preprint arXiv:2205.01666, 2022.

---

### Author Response · Authors · 2022-11-14
**Common Response**

We sincerely thank all reviewers for constructive suggestions and recognition to our work. We are encouraged that reviewers find our proposed method "interesting and effective" (Reviewer eUx6) and "well-designed" (Reviewer bCYZ); our results "of high quality" (Reviewer eUx6), "compelling" (Reviewer zbyZ) and "promising" (Reviewer bCYZ). We have posted a common response post for the issue of novelty of the compositional representation and separate response posts for each reviewer.

We have also updated our submission to include the following changes according to reviewers' feedback. Note that all revisions in the main paper and supplementary material are highlighted in blue.

1. Missing references are added in Section 2 of the main paper.

2. Discussions on the compositional representation and its application in NeRF are included in Section 2 of the main paper.

3. Clarifications on how FID/KID are calculated are added to Section 4.1 of the main paper.

4. Discussions on the "line artifacts" are included in main paper Section 5 "Limitations" 3 and supplementary material Section 4.1.

5. Comparison with 2D human generation and related discussions are added in main paper Section 5 "Limitations" 4.

6. Comparison of interpolation and inversion with baseline is added in supplementary material Section 4.2 and Figure 5.

Please do not hesitate to let us know if you have any additional comments or there are more clarifications that we can offer.

---

### Decision · Program_Chairs · 2023-01-20

**Decision:**

Accept: notable-top-25%

**Justification For Why Not Higher Score:**

The paper has received positive scores from all reviewers.

The originality has been criticized but not really an issue. Therefore, the meta-reviewer recommends this paper to be accepted as a spotlight.

**Justification For Why Not Lower Score:**

All reviewers are positive about the work.

**Metareview: Summary, Strengths And Weaknesses:**

The work proposes to build a generative model of human body by composing NeRF of parts, which is an articulated 3D shape. All reviewers are positive about the performance of the work. There are minor concerns regarding the originality of the work. The meta-reviewer agrees with the comments of all reviewers and supports to accept the work.

**Note From Pc:**

if the above contains the word "oral" or "spotlight" please see: "oral" presentation means -> notable-top-5% and "spotlight" means -> notable-top-25%. As stated in our emails, we are disassociating presentation type from AC recommendations

**Summary Of Ac-Reviewer Meeting:**

Not a borderline paper.